# Salicylic Acid Improves Antioxidant Defense System and Photosynthetic Performance in *Aristotelia chilensis* Plants Subjected to Moderate Drought Stress

**DOI:** 10.3390/plants11050639

**Published:** 2022-02-26

**Authors:** Jorge González-Villagra, Marjorie M. Reyes-Díaz, Ricardo Tighe-Neira, Claudio Inostroza-Blancheteau, Ana Luengo Escobar, León A. Bravo

**Affiliations:** 1Departamento de Ciencias Agropecuarias y Acuícolas, Facultad de Recursos Naturales, Universidad Católica de Temuco, Temuco 4781312, Chile; jorge.gonzalez@uct.cl (J.G.-V.); rtighe@uct.cl (R.T.-N.); claudio.inostroza@uct.cl (C.I.-B.); 2Núcleo de Investigación en Producción Alimentaria, Facultad de Recursos Naturales, Universidad Católica de Temuco, Temuco 4781312, Chile; 3Departamento de Ciencias Químicas y Recursos Naturales, Facultad de Ingeniería y Ciencias, Universidad de La Frontera, Temuco 4811230, Chile; marjorie.reyes@ufrontera.cl; 4Center of Plant, Soil Interaction and Natural Resources Biotechnology, Scientific and Technological Bioresource Nucleus (BIOREN), Universidad de La Frontera, Temuco 4811230, Chile; ana.luengo@ufrontera.cl; 5Departamento de Ciencias Agronómicas y Recursos Naturales, Facultad de Ciencias Agropecuarias y Forestales, Universidad de La Frontera, Temuco 4811230, Chile

**Keywords:** superoxide dismutase activity, ascorbate peroxidase activity, CO_2_ assimilation, total phenolics, plant growth

## Abstract

Salicylic acid (SA) has been shown to ameliorate drought stress. However, physiological and biochemical mechanisms involved in drought stress tolerance induced by SA in plants have not been well understood. Thus, this study aimed to study the role of SA application on enzymatic and non-enzymatic antioxidants, photosynthetic performance, and plant growth in *A. chilensis* plants subjected to moderate drought stress. One-year-old *A. chilensis* plants were subjected to 100% and 60% of field capacity. When plants reached moderate drought stress (average of stem water potential of −1.0 MPa, considered as moderate drought stress), a single SA application was performed on plants. Then, physiological and biochemical features were determined at different times during 14 days. Our study showed that SA application increased 13.5% plant growth and recovered 41.9% *A_N_* and 40.7% *g_s_* in drought-stressed plants on day 3 compared to drought-stressed plants without SA application. Interestingly, SOD and APX activities were increased 85% and 60%, respectively, in drought-stressed SA-treated plants on day 3. Likewise, SA improved 30% total phenolic content and 60% antioxidant capacity in drought-stressed *A. chilensis* plants. Our study provides insight into the SA mechanism to tolerate moderate drought stress in *A. chilensis* plants.

## 1. Introduction

Drought has become one of the most important environmental stress factors for plants, reducing yields by more than 50% [1,2,3,4]. This stress negatively affects carbon accumulation, tissue expansion, and plant development [5,6,7]. Stomatal closure is among the first physiological processes affected by drought stress, decreasing photosynthesis and plant growth [8,9,10,11]. Although it has been reported that photosynthesis is mainly inhibited by stomatal closure, it might be also affected by metabolic impairment. Thus, studies have reported that drought stress decreases ribulose-1,5-bifosfate carboxylase/oxygenase (Rubisco) activity, Calvin and Benson cycle enzymes inactivation, low adenosine triphosphate (ATP), and damage to photosystem II [8,12,13,14,15]. However, the mechanisms by which metabolic impairment causes photosynthesis inhibition under drought stress have not been fully elucidated [16,17]. Plants subjected to drought stress increase reactive oxygen species (ROS) production in different cellular organelles such as chloroplasts, mitochondria, and peroxisomes, generating oxidative stress, damaging DNA, proteins, and lipids [18,19,20]. Thus, González-Villagra et al. [21] showed that oxidative stress increased about 50% after 20 days of drought stress compared to well-watered *Aristotelia chilensis* plants under greenhouse conditions.

On the other hand, plants have developed morphological, physiological, and molecular mechanisms to tolerate drought stress. Stomatal closure is among the most important physiological mechanisms to tolerate drought stress by preventing water loss by transpiration [22]. Other important physiological mechanisms to tolerate drought stress are the biosynthesis of enzymatic and non-enzymatic antioxidants [23,24,25,26]. The primary defense line of antioxidant enzymes is superoxide dismutase (SOD) and ascorbate peroxidase (APX), which scavenge ROS such as superoxide anions (O_2_^.^) and hydrogen peroxide (H_2_O_2_), forming water and oxygen [27,28,29]. Among non-enzymatic antioxidants, phenolic compounds are highly reactive as hydrogen or electron donors and have the ability to chelate transition metal ions, inhibit lipoxygenase activity, and scavenge ROS, which is considered as an adaptive response to drought stress [30,31,32,33].

Among plant hormones, salicylic acid (SA) is involved in plant defense against microbial pathogens, activating the systemic acquired resistance (SAR) [34,35]. Under non-stressed conditions, SA plays an important role in nutrient uptake, cell elongation, photosynthetic pigment levels, photosynthetic activity, and source-to-sink regulation, regulating plant growth and development [36,37,38,39]. Currently, SA has been suggested as an important molecule involved in plant tolerance to drought stress [38,40]. For example, Chen et al. [41] showed that SA application increased CO_2_ assimilation and plant growth in drought-stressed *Zoysia japonica* plants. Currently, Zafar et al. [42] reported that leaf gas exchange, SOD and APX activities, and dry weight accumulation were improved by SA application in *Conocarpus erectus* and *Populus deltoides* plants subjected to drought stress. Therefore, SA is a critical plant hormone that regulates the activation of biotic and abiotic stress defense. However, the physiological and biochemical mechanisms involved in drought stress tolerance induced by SA in plants have not been well understood [42,43].

*Aristotelia chilensis* (Mol.), also known as “maqui”, is an important endemic berry that grows in Southern Chile [26,44,45,46,47,48]. Currently, several studies related to crop management and morphological and physiological characterization have been performed in this species considering their antioxidant properties [21,47,48,49,50,51].

Therefore, we hypothesized that SA application triggers enzymatic and non-enzymatic antioxidant mechanisms, which ameliorate oxidative stress, improving photosynthesis performance and plant growth of *A. chilensis* plants subjected to moderate drought stress. Thus, the aim of this study was to study the role of SA application on enzymatic and non-enzymatic antioxidants, photosynthetic performance, and plant growth in *A. chilensis* plants subjected to moderate drought stress. Our results provide new knowledge into the role of SA on the antioxidant defense system in *A. chilensis* to tolerate moderate drought stress.

## 2. Results

### 2.1. Relative Growth Rate and Plant Water Status in A. chilensis

Our results revealed that moderate drought stress negatively affected plant growth in *A. chilensis* plants, reducing about 27% relative growth rate (RGR) compared to well-irrigated plants (control, 100% FC) at days 3 and 7 of the experiment (Table 1). However, SA application stimulated plant growth in moderate drought-stressed *A. chilensis* plants, increasing 13.5%, 7.8%, and 5.4% RGR with respect to moderate drought-stressed plants without SA-treated plants at days 3, 7, and 14, respectively. A similar tendency was observed in well-irrigated plants, where RGR was increased 10.6% and 15% at days 3 and 7 of the experiment in plants with SA application, showing no differences at day 14 of the experiment with respect to plants without SA application. In our study, plants subjected to moderate drought stress showed significantly lower stem water potential (Ψ_w_) (around −1.3 MPa) compared to well-irrigated plants at day 3 of the experiment (−0.5 MPa) (Figure 1A). Interestingly, when SA application was performed, Ψ_w_ increased in moderate drought-stressed plants, reaching well-irrigated plant levels at day 7. However, on days 7 and 14 of the experiment, Ψ_w_ in moderate drought-stressed plants with SA application were reduced, reaching similar values as drought-stressed plants without SA application. In contrast, no changes were observed in well-irrigated plants throughout the experiment. Concerning relative water content (RWC), moderate drought-stressed plants showed slightly lower levels (6%) than well-irrigated plants (Figure 1B). Meanwhile, moderate drought-stressed plants with SA application showed similar RWC values compared to plants subjected to drought stress without SA application. With respect to control plants (100% FC) with and without exogenous SA, no variations were shown in RWC during the experiment (Figure 1B).

### 2.2. Photosynthetic Performance in A. chilensis Subjected to Moderate Drought Stress

We observed a significant interaction among time, irrigation treatments, and SA levels for electron transport rate (ETR), effective quantum yield (ΦPSII), CO_2_ assimilation, stomatal conductance, and transpiration (*p* ≤ 0.05), except WUE*i*. In our study, moderate drought-stressed plants had similar levels as well-irrigated plants in terms of ΦPSII and ETR, remaining unchanged during the experiment (Figure 2). Surprisingly, exogenous SA application increased 41% ΦPSII and 33% ETR in plants subjected to moderate drought stress at day 3, recovering values at days 7 and 14 of the experiment compared to drought-stressed plants without SA application, showing a transitory improvement of this parameter. A similar tendency showed SA-treated well-irrigated plants, where ΦPSII and ETR were increased at day 3 and 7, declining at day 14. On the other hand, moderate drought stress negatively affected net CO_2_ assimilation (*A_N_*) in *A. chilensis* plants, showing 30.8% lower *A_N_* at the beginning of the experiment compared with plants subjected to 100% FC (Figure 3A). SA application recovered and increased *A_N_* in moderate drought-stressed plants at day 3, being 41.9% higher than moderate drought-stressed plants without SA application. However, *A_N_* decreased in moderate drought-stressed plants treated with SA at days 7 and 14, reaching levels of plants without SA. A similar tendency was observed in 100% FC plants where exogenous SA increased 33.8% *A_N_* compared with plants without SA at day 3; however, *A_N_* reached control values from day 7 of the experiment, reaching 100% FC without SA. In our study, lower stomatal conductance (*g_s_*) levels were observed in moderate drought-stressed plants than well-irrigated ones (Figure 3B). Exogenous SA increased 41% *g_s_* in moderately stressed plants compared to those without SA at day 3, decreasing at days 7 and 14. Well-irrigated SA-treated plants progressively increased *g_s_* from day 7 and 14 compared to plants without SA treatment. Concerning transpiration (*E*), this parameter was significantly reduced in plants (60% FC) throughout the experiment compared to well-irrigated plants (Figure 3C). A similar tendency was observed when SA was applied to plants, where moderate drought-stressed plants increased *E* levels compared to non-SA-treated plants at day 3 of the experiment, while well-irrigated SA-treated plants showed an increase in *E* levels at days 7 and 14 of the experiment. Furthermore, we observed that SA application improved WUE*i* in moderate drought-stressed plants at day 3, reaching the highest level at day 7 of the experiment, while well-irrigated plants had a slight increase in WUE*i* only at day 3 (Figure 3D).

### 2.3. Antioxidant Capacity and Total Phenolic Content Determination

In our study, antioxidant capacity (AC) and total phenolic content (TPC) showed statistically significant interaction among day, irrigation treatment, and SA treatment (*p* ≤ 0.05). Our results showed that moderate drought-stressed plants had lower antioxidant capacity (AC) compared to well-irrigated plants at day 3, unchanging until day 14 of the experiment, where a slight increase (about 12%) was observed (Figure 4A). However, SA application significantly increased (around 30%) AC in moderate drought-stressed plants on days 7 and 14 of the study. Likewise, well-irrigated plants showed a slightly higher AC level at day 7 when exogenous SA was applied; however, their AC levels decreased at day 14 of the experiment. In TPC, plants subjected to moderate drought stress had similar levels as well-irrigated plants during the experiment, except day 0 (Figure 4B). Interestingly, exogenous SA application significantly improved TPC levels (about 60%) at days 3 and 7 in plants subjected to drought stress, while well-irrigated plants showed the highest TPC levels at day 3, which was 2-fold greater than non-SA-treated plants (Figure 4B).

### 2.4. Lipid Peroxidation

As an indicator of oxidative stress, lipid peroxidation was determined in *A. chilensis* plants. Our results showed that lipid peroxidation was higher in moderate drought-stressed *A. chilensis* plants, where a significant increase (around 2.5-fold) compared to well-irrigated plants was observed throughout the experiment (Figure 4C). Meanwhile, drought-stressed plants subjected to SA application significantly decreased their MDA content from day 7, reaching the lowest levels at day 14. On the other hand, well-irrigated SA-treated plants remained unchanged in their MDA content during the experiment, showing a mean value of 17 nmol g^−1^ fresh weight.

### 2.5. Superoxide Dismutase and Ascorbate Peroxidase Activity in A. chilensis

Superoxide dismutase (SOD) and ascorbate peroxidase (APX) activities were determined as an enzymatic antioxidant defense mechanism in *A. chilensis*. We observed a significant interaction among time, irrigation treatments, and SA levels for SOD and APX activities (*p* ≤ 0.05). Plants subjected to drought stress showed an evident increment in SOD activity compared to well-irrigated *A. chilensis* plants throughout the experiment (Figure 5A). Meanwhile, well-irrigated plants maintained relatively constant SOD activity in our study. Exogenous SA application significantly increased SOD activity in moderate drought-stressed and well-irrigated plants. The highest SOD activity was observed in drought-stressed SA-treated plants at day 3, where it increased about 85% with respect to drought-stressed plants without SA application (Figure 5A). However, SOD activity decayed in SA-treated plants subjected to drought, reaching the same levels as non-SA-treated plants at day 14 of the experiment. Concerning APX, we observed that drought-stressed plants had significantly higher levels than well-irrigated plants, exhibiting around 2-fold greater activity throughout the experiment (Figure 5B). Interestingly, exogenous SA application increased (around 60%) APX activity in moderate drought-stressed *A. chilensis* plants at day 3. Then, APX activity decreased to reach similar levels as non-SA-treated plants at day 14 of the experiment. Similar behavior was observed in well-irrigated plants, where APX activity slightly increased from day 3, reaching the highest level at day 7, around 30% greater than well-irrigated plants without SA application (Figure 5B).

## 3. Discussion

Drought stress is the most severe manifestation of climate change, causing a severe threat to food security [52,53]. The results from our study revealed that the relative growth rate was decreased by 27% in *A. chilensis* plants subjected to moderate drought stress compared to well-irrigated plants at days 3 and 7 of the experiment (Table 1). Likewise, plant water status, including stem water potential (Ψ_w_) and relative water content (RWC), was reduced concomitantly with plant growth in moderate drought-stressed *A. chilensis* plants (Figure 1). In fact, Ψ_w_ reached a stable lowest value of −1.3 MPa in drought-stressed plants (irrigation at 60% FC), while well-irrigated plants maintained their Ψ_w_ around −0.5 MPa during the experiment. Similarly, moderate drought-stressed *A. chilensis* plants had a slightly lower (6%) RWC level compared to well-irrigated plants. The negative effects of moderate drought stress on plant growth and plant water status have been reported for several species such as *Phaseolus vulgaris* [54], *Vaccinium corymbosum* [55], *Malus domestica* [56], and *Punica granatum* [57]. We previously reported that severe drought stress reduced 5-fold Ψ_w_ and 71% plant growth in *A. chilensis* plants after 20 days of water restriction [21]. In this context, it is well-known that plants induce stomatal closure, which is among the earliest response to drought stress, as a mechanism for preventing water loss, decreasing CO_2_ assimilation (*A_N_*), and inhibiting plant growth [58,59,60]. Here, we observed that *A_N_* and *g_s_* were significantly decreased by 30.8% and 21.4%, respectively, in plants subjected to drought stress contrasted with well-irrigated plants, which exhibited the highest and steady *A_N_* and *g_s_* values throughout the experiment (Figure 3). Thus, plant growth inhibition could be explained due to lower *g_s_* and *A_N_* levels in drought-stressed *A. chilensis* plants in our experiment. Nonetheless, photosynthesis might be also affected by metabolic impairment. Thus, it has been reported that reactive oxygen species (ROS) are produced in different cellular compartments such as chloroplasts, mitochondria, and peroxisomes [19,61]. ROS production in chloroplasts is mediated by photoreduction of O_2_ and the ground-state oxygen to singlet state in the reaction centers of Photosystem I (PSI) and Photosystem II (PSII), and then generating different types of ROS such as ^1^O_2_, O_2_.-, H_2_O_2_, and OH^.^, decreasing photosynthetic performance [62]. Under drought stress, ROS production induces damage to DNA, proteins, carbohydrates, and lipids, triggering oxidative stress [18]. Thus, lipid peroxidation was measured as an indicator of oxidative stress in our study (Figure 4C). Our results showed a significant increase (around 2.5-fold) in lipid peroxidation in moderate drought-stressed *A. chilensis* plants, which was similar to our previous study in the same species [21]. Therefore, *A. chilensis* plants were biochemically and physiologically affected by drought stress, increasing lipid peroxidation and decreasing *g_s_*, *A_N_*, and plant growth.

Salicylic acid (SA) is a plant hormone regulating plant growth and development under non-stressed conditions [37,38,39]. Currently, SA has been suggested as an important molecule involved in ameliorating plant drought stress [38,39,40,63]. However, the physiological and biochemical mechanisms involved on drought stress tolerance induced by SA in plants have not been well understood [42,43]. Some reports have shown that SA improves SOD and APX activities and leaf gas exchange in *Conocarpus erectus*, *Populus deltoids,* and *Olea europaea* plants subjected to drought stress [38,42]. Therefore, we hypothesized that SA application triggers enzymatic and non-enzymatic antioxidant mechanisms, ameliorating oxidative stress and improving photosynthesis performance and plant growth in *A. chilensis* plants subjected to drought stress. We observed that three days post SA application, *A_N_* recovered in drought-stressed plants compared to drought-stressed plants without SA application (Figure 3A). A similar tendency was observed in well-watered plants where exogenous SA increased *A_N_* three days post application compared with plants without SA. Concomitantly, exogenous SA increased *g_s_* in stressed plants compared to those without SA (Figure 3B). Interestingly, we observed that SA application improved WUE*i* in drought-stressed plants at day 3, reaching the highest level at day 7 of the experiment (Figure 3D). In addition, we observed that higher CO_2_ assimilation in SA-treated plants was concomitant with greater plant growth and better plant water status in *A. chilensis* plants subjected to drought stress in our experiment. Thus, SA application recovered plant growth in drought-stressed *A. chilensis* plants, with respect to drought-stressed plants without SA treatment (Table 1). Our results agree with Chen et al. [39], where they showed that SA application increased CO_2_ assimilation and plant growth in drought-stressed *Zoysia japonica* plants. Recently, Shemi et al. [64] showed that SA treatment increased about 25% plant growth and crop yield in *Zea mays* plants subjected to drought stress due to higher CO_2_ assimilation and stomatal conductance. Interestingly, the authors also showed that the intercellular CO_2_ concentration remained unchanged between SA-treated and non-SA-treated plants, which could indicate a better metabolic efficiency triggered by SA. We observed that exogenous SA application increased electron transport rate (ETR) and effective quantum yield (ΦPSII) in plants subjected to moderate drought stress at day 3 of the experiment (Figure 3). In fact, Khalvandi et al. [65] suggested that SA may also regulate some physiological responses related to carbon uptake and/or fixation in the chloroplasts, such as high ribulose-1,5-bifosfate carboxylase/oxygenase (Rubisco) concentration and activity, increasing CO_2_ assimilation and plant growth. Likewise, Shao et al. [66] showed that Rubisco and Rubisco activase enzyme activities increased due to SA application in *Zea mays* plants subjected to drought stress. The authors also reported that SA induced high transcript levels of Rubisco large subunit (Rbc L), α-form Ribulose-1,5-bisphosphate carboxylase/oxygenase (ZmRCAα), and β-form Ribulose-1,5-bisphosphate carboxylase/oxygenase (ZmRCAβ) mRNA, which may confirm the positive effects of SA on CO_2_ assimilation in our study. It has been reported that SA induces stomatal closure under biotic stress, preventing pathogen invasion into plants [35]. However, we observed a stomatal aperture in SA-treated *A. chilensis* plants, where *g_s_* increased 41% in moderate drought-stressed plants compared to moderate drought-stressed plants without SA at day 3 (Figure 3B). Our results were similar to those reported by Shemi et al. [64] and Habibi et al. [67], where they showed that SA induced stomatal aperture in *Z. mays* and *Hordeum vulgare* plants subjected to drought stress. Recently, Zamora et al. [68] showed that *Arabidopsis* and *Solanum lycopersicum* plants did not reduce stomatal aperture after SA spraying. The authors also suggested that SA-induced stomatal closure apparently requires high concentrations or prolonged SA treatments. Therefore, we can suggest that SA-induced stomatal closure seems dependent of the species, SA concentration, and/or biotic or abiotic stress specific. On the other hand, Nazar et al. [37] also suggested that SA helps to maintain the integrity of light-harvesting apparatus by enzymatic and non-enzymatic antioxidants, being an important mechanism for increasing photosynthesis under drought stress. In our study, we observed that SA application had a transitory stimulus of photosynthetic performance including ETR, ΦPSII, *g_s_*, *A_N_*, and plant growth at day 3 post SA application. It has been reported that SA may undergo different chemical modifications such as glycosylation, methylation, and amino acid conjugation, metabolizing into inactive or storage forms to modulate its activity and fine-tune its levels [69,70]. Thus, the transitory improvement of photosynthetic performance and plant growth in *A. chilensis* plants could be explained by SA metabolism, which modulate its activity and levels in plants.

As we mentioned before, plants have developed complex mechanisms to scavenge ROS and avoid oxidative stress under drought stress, such as enzymatic and non-enzymatic antioxidants [24,25,26]. Thus, SOD and APX activities were determined as an enzymatic antioxidant defense mechanism in *A. chilensis*. In our study, exogenous SA application significantly increased (85%) SOD activity in drought-stressed plants at day 3 with respect to drought-stressed plants without SA application (Figure 5A). However, SOD activity in SA-treated plants reached levels of control (non-SA-treated plants) on day 14 of the experiment. The same tendency showed APX activity, where SA application increased (around 60%) its levels in drought-stressed *A. chilensis* plants at day 3, decreasing to reach similar levels as non-SA-treated plants at day 14 of the experiment (Figure 5B). Similar results were reported by Mohi-Ud-Din et al. [63] and Sharma et al. [71], where SA application increased SOD and APX activities in *Glycine max* and *Phaseolus vulgaris* plants subjected to drought stress. Wang et al. [72] showed that *Hordeum vulgare* plants overexpressing the *isochorismate synthase* gene, which is a key gene for SA biosynthesis and which significantly increased SOD and APX activities, decreasing ROS and lipid peroxidation induced by drought stress, could partly explain our results. In addition, as a non-enzymatic antioxidant, exogenous SA application significantly increased TPC levels (60%) at days 3 and 7 in plants subjected to drought stress, while the same treatment increased 30% AC in drought-stressed plants at days 7 and 14 of the study (Figure 4). Phenolic compounds are important molecules to counteract oxidative stress, which are highly reactive as hydrogen or electron donors and have the ability to chelate transition metal ions, scavenging ROS, are being considered as an adaptive response to drought stress [30,31,32,33]. Interestingly, we observed that drought-stressed *A. chilensis* plants subjected to SA application significantly decreased their lipid peroxidation, measured as MDA content. This result suggests that both enzymatic and non-enzymatic antioxidant mechanisms could contribute to coping with ROS, decreasing oxidative stress, and improving photosynthetic performance and plant growth in *A. chilensis* plants subjected to drought stress. Thus, a schematic representation is shown in Figure 6, where we suggested that SA induces enzymatic and non-enzymatic antioxidant mechanisms to tolerate drought stress, decrease oxidative stress, and improve photosynthesis performance and plant growth in *A. chilensis* plants. However, SA molecular regulation mechanisms to induce enzymatic and non-enzymatic antioxidant mechanisms in *A. chilensis* plants under drought stress is still unknown.

## 4. Materials and Methods

### 4.1. Plant Material and Experimental Conditions

One-year-old *A*. *chilensis* plants obtained in vitro conditions were used in this study, which Plangen Co., Máfil, Chile donated. Plants of uniform size were conditioned in plastic pots containing 1.5 L of Andisol soil under greenhouse conditions for 2 weeks according to González-Villagra et al. [21,26]. Greenhouse conditions were a 16/8 h light/dark photoperiod, 23 ± 2 °C temperature, 70% relative humidity, and a mean 300 µmol photons m^−2^ s^−1^. Plants were acclimated for two weeks. Then, plants were divided into two groups; plants daily irrigated (DI), maintained at 100% of field capacity (100% FC) and plants non-irrigated (NI), maintained at 60% of FC reaching an average stem water potential of −1.0 MPa after 10 days of treatment (considered as moderate drought stress for this species, according to our previous study; González-Villagra et al. [21,26]). When NI plants reached moderate drought stress, a single application of 0.5 mM salicylic acid (SA) (Sigma, St. Louis, MO, USA) was performed, spraying homogeneously for both irrigation treatments (+SA) [73,74]. The SA was dissolved in ultrapure water containing 0.05% (*v*/*v*) of Tween 20 used as the surfactant wetting agent. Control solution only contained ultrapure water with Tween 20 (-SA). The experiment was carried out for 14 days. At different time points post SA application (0, 3, 7, and 14 days), in vivo gas-exchange determination was performed, and leaf samples were collected in the middle of the light period, frozen in liquid nitrogen, and stored at −80 °C until biochemical analyses.

### 4.2. Plant Growth and Water Status Measurements in A. chilensis

#### 4.2.1. Relative Growth Rate (RGR)

In order to measure growth rate during the experiment, the Hoffmann and Poorter [75] protocol was used. It was calculated by relative growth rate (RGR) from the mean natural logarithm-transformed dry weight (DW). The t1 was time 0 and t2 were the times 3, 7, and 14 days. RGR was calculated by Formula (1).
RGR = [(lnDW2) − (lnDW1)/(t2 − t1)](1)

#### 4.2.2. Plant Water Status

Stem water potential (Ψ_w_) was measured between 08:00 and 10:00 on the leaf petiole using a Scholander chamber Model 1000 (PMS, Instruments Co., Corvallis, OR, USA) based on the Begg and Turner [76] protocol. For this, 90 min before measurement, leaves were covered with aluminum foil in a plastic bag. In addition, relative water content (RWC) was determined following the Rahimi et al. [77] method. Two leaves were removed, weighed, and immersed into double distilled water for the next 24 h at 4 °C in dark conditions. Then, leaves were oven-dried to a constant weight at 60 °C. Formula (2) was used to determine RWC:RWC = [(fresh weight − dry weight)/(turgid weight − dry weight)] × 100(2)

### 4.3. Photosynthetic Performance

Electron transport rate (ETR), effective quantum yield (ΦPSII), CO_2_ assimilation (*A_N_*), stomatal conductance (*g_s_*), and transpiration (*E*) were measured in order to determine the photosynthetic performance of *A. chilensis* plants. Thus, in vivo measurements were performed using a portable infrared CO_2_ analyzer equipped with a Li-Cor LR6400 cuvette measurement with its light source (LI-COR Inc., Lincoln, NE, USA), during the light period (08:00 to 10:00 h), as described by Reyes-Díaz et al. [78]. The portable photosynthesis system controlled the light source (400 µmol photons m^−2^ s^−1^), temperature (25 °C), humidity, and CO_2_ concentration. External air with CO_2_ was used to obtain a concentration reference of 400 µmol mol^−1^, with a flow rate of 300 mL min^−1^ and 80% external relative humidity. Four measurements per plant were performed.

### 4.4. Lipid Peroxidation

Leaf samples were macerated with a mixture of trichloroacetic acid (TCA) and thiobarbituric acid (TBA). Then, the homogenate was centrifuged at 13,000 rpm for 10 min at 4 °C. The supernatant was collected and used for lipid peroxidation (LP) determination. The LP was measured based on the formation of thiobarbituric acid-reactive substances (TBARS) according to Du and Bramalage [79]. In order to correct the interference generated by TBARS-sugars complexes, absorbance was measured at 440, 532, and 600 nm (UV/VIS Unico SpectroQuest 2800). The TBARS content was expressed as nmol of malondialdehyde (MDA) per gram of fresh weight (nmol MDA g^−1^ FW). Formula (3) was used to determine the content of TBARS in terms of MDA equivalent:MDA equivalent (nmol/mL) = [(*A*_532_ − *A*_600_)/157,000] × 10^6^
(3)

### 4.5. Antioxidant Capacity and Total Phenolic Content Determination

Leaf samples were ground in liquid nitrogen and macerated with ethanol (80% *v*/*v*). The homogenate was centrifuged at 13,000 rpm for 10 min at 4 °C. The supernatant was collected and used for antioxidant capacity and total phenolic content determinations. Antioxidant capacity was determined by the DPPH (2.2-diphenyl-1-picryl-hydrazyl) assay [80], measuring the absorbance at 515 nm (UV/VIS Unico SpectroQuest 2800). Antioxidant capacity was expressed as mg of Trolox equivalents per gram of fresh weight (mg TE g^−1^ FW). Total phenolic content was determined using the Folin–Ciocalteu method (Singleton and Rossi [81]), measuring the absorbance at 765 nm (UV/VIS Unico SpectroQuest 2800) using caffeic acid as standard. Total phenolic content was expressed as mg of caffeic acid equivalents per gram of fresh weight (mg CAE g^−1^ FW).

### 4.6. Superoxide Dismutase (SOD) and Ascorbate Peroxidase (APX) Activities in A. chilensis

Leaf samples were ground in liquid nitrogen and macerated with 50 mM potassium phosphate buffer (K_2_HPO_4_–KH_2_PO_4_, 50 mM, pH 7.0). Then, the homogenate was centrifuged at 11,000 g for 15 min at 4 °C, and the supernatant (crude extract) was used for SOD and APX determinations. Superoxide dismutase (SOD) (EC. 1.15.1.1) activity was determined by measuring the photochemical inhibition of nitroblue tetrazolium (NBT) as reported by Giannopolitis et al. [82]. Briefly, crude extract (20 µL) was added to a reaction mixture containing potassium phosphate buffer, 0.1 mM ethylenediaminetetraacetic acid (EDTA), 13 mM methionine, and 322 µM NBT. Riboflavin was added to start the reaction. Reaction mixtures were illuminated for 15 min, and the absorbance was measured at 560 nm. One SOD unit was defined as the amount of enzyme corresponding to 50% inhibition of the NBT reduction (Donahue et al. [83]). The enzyme activity was standardized for the protein content. Protein in the crude enzyme extract was determined by the Bradford method (Bradford [84]).

Ascorbate peroxidase (APX) (EC. 1.11.1.11) activity was determined by Nakano and Asada [85]. For this, the crude extract (40 µL) was diluted in a reaction mixture containing 1 mL of extraction buffer, 5 µL of H_2_O_2_ (30% *v*/*v*), and 40 µL of 10 mM ascorbic acid. Enzyme activity was calculated using a molar extinction coefficient of 2.8 mM cm^−1^.

### 4.7. Experimental Design and Statistical Analyses

A completely randomized design was used with three replicates for each treatment and time. Data were tested for homogeneity and normality of variances using the Levene and Kolmogorov–Smirnov tests, respectively. Then, data were analyzed using three-way ANOVA, where factors were drought treatments, SA application, and time post SA application. Tukey’s multiple comparison test *p* ≤ 0.05 was used. Sigma Stat v.2.0 (SPSS, Chicago, IL, USA) was used to perform the statistical analysis.

## 5. Conclusions

Our study demonstrated that moderate drought stress negatively affects the physiological and biochemical features of *A. chilensis* plants. Our study provides insight into the SA mechanism to tolerate drought stress in *A. chilensis* plants. Our results showed that SA application increases SOD and APX activities, total phenolic content, and antioxidant capacity concomitant with decreased oxidative stress, and improved photosynthesis performance and plant growth in *A. chilensis* plants subjected to moderate drought stress. However, further molecular studies are needed to understand the SA mechanism to improve moderate drought stress tolerance in *A. chilensis*.

## Figures and Tables

**Figure 1 plants-11-00639-f001:**
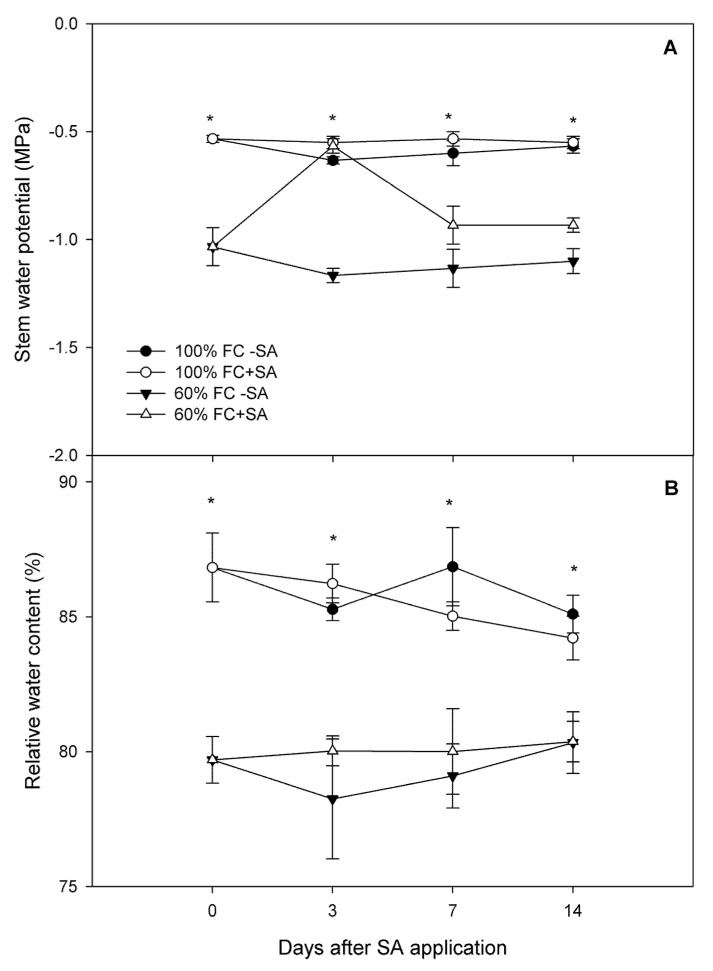
(**A**) Stem water potential and (**B**) relative water content in *A. chilensis* plants grown under 100% and 60% FC and two SA doses (0 and 0.5 mM) at different times post SA application. The bars are means ± SE (*n* = 3). According to Tukey’s test, asterisks indicate significant differences between irrigation treatments of the experiment for the same SA treatment and time post SA application (*p* ≤ 0.05).

**Figure 2 plants-11-00639-f002:**
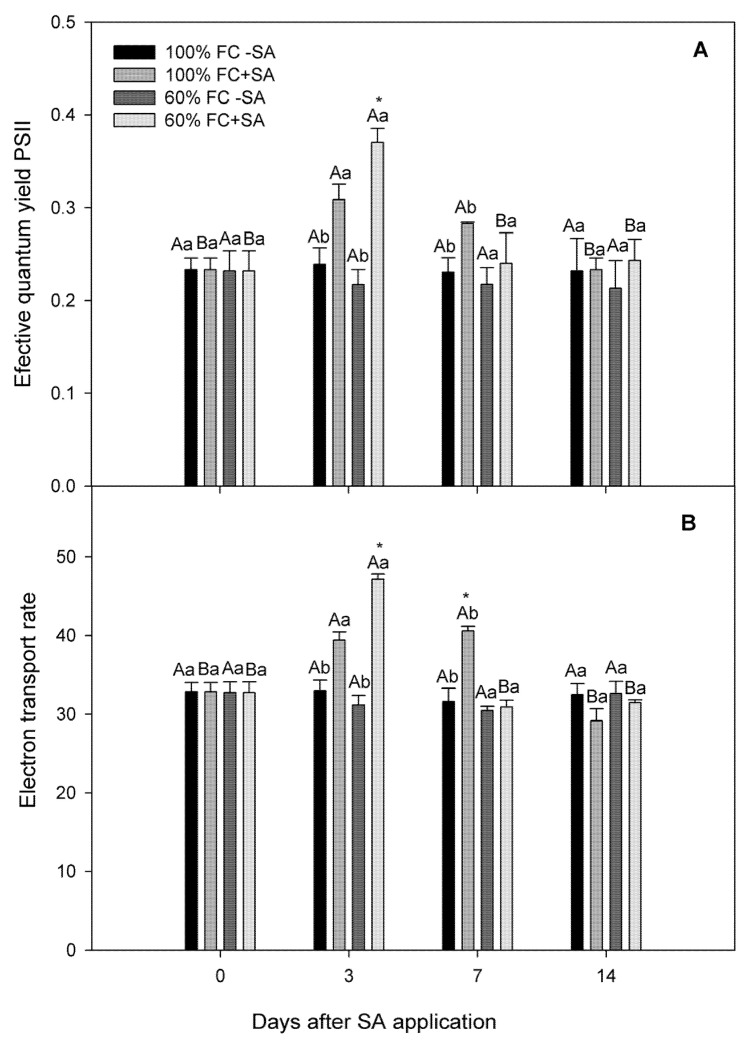
Photochemical parameters in *A. chilensis* plants grown under 100% and 60% FC and two SA doses (0 and 0.5 mM) at different times. (**A**) effective quantum yield (ΦPSII) and (**B**) electron transport rate (ETR). The bars are means ± SE (*n* = 3). Different uppercase letters indicate significant differences among days for the same SA and irrigation treatment according to Tukey’s test (*p* ≤ 0.05). Different lowercase letters indicate significant differences between SA treatment for the same irrigation treatment and time post SA application according to Tukey’s test (*p* ≤ 0.05). According to Tukey’s test, asterisks indicate significant differences between irrigation treatments of the experiment for the same SA treatment and time post SA application (*p* ≤ 0.05).

**Figure 3 plants-11-00639-f003:**
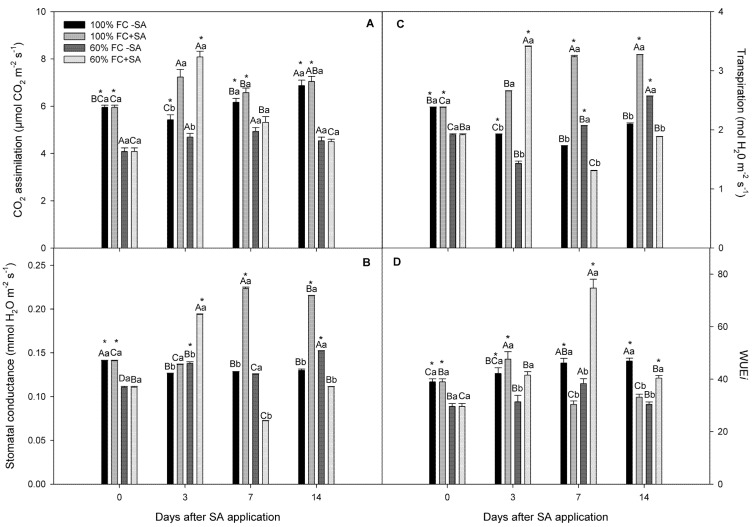
Gas-exchange measurements in *A. chilensis* plants grown under 100% and 60% FC and two SA doses (0 and 0.5 mM) at different times. (**A**) CO_2_ assimilation, (**B**) stomatal conductance, (**C**) transpiration, and (**D**) intrinsic water-use efficiency (WUE*_i_*). The bars are means ± SE (*n* = 3). Different uppercase letters indicate significant differences among days for the same SA and irrigation treatment according to Tukey’s test (*p* ≤ 0.05). Different lowercase letters indicate significant differences between SA treatment for the same irrigation treatment and time post SA application according to Tukey’s test (*p* ≤ 0.05). According to Tukey’s test, asterisks indicate significant differences between irrigation treatments of the experiment for the same SA treatment and time post SA application (*p* ≤ 0.05).

**Figure 4 plants-11-00639-f004:**
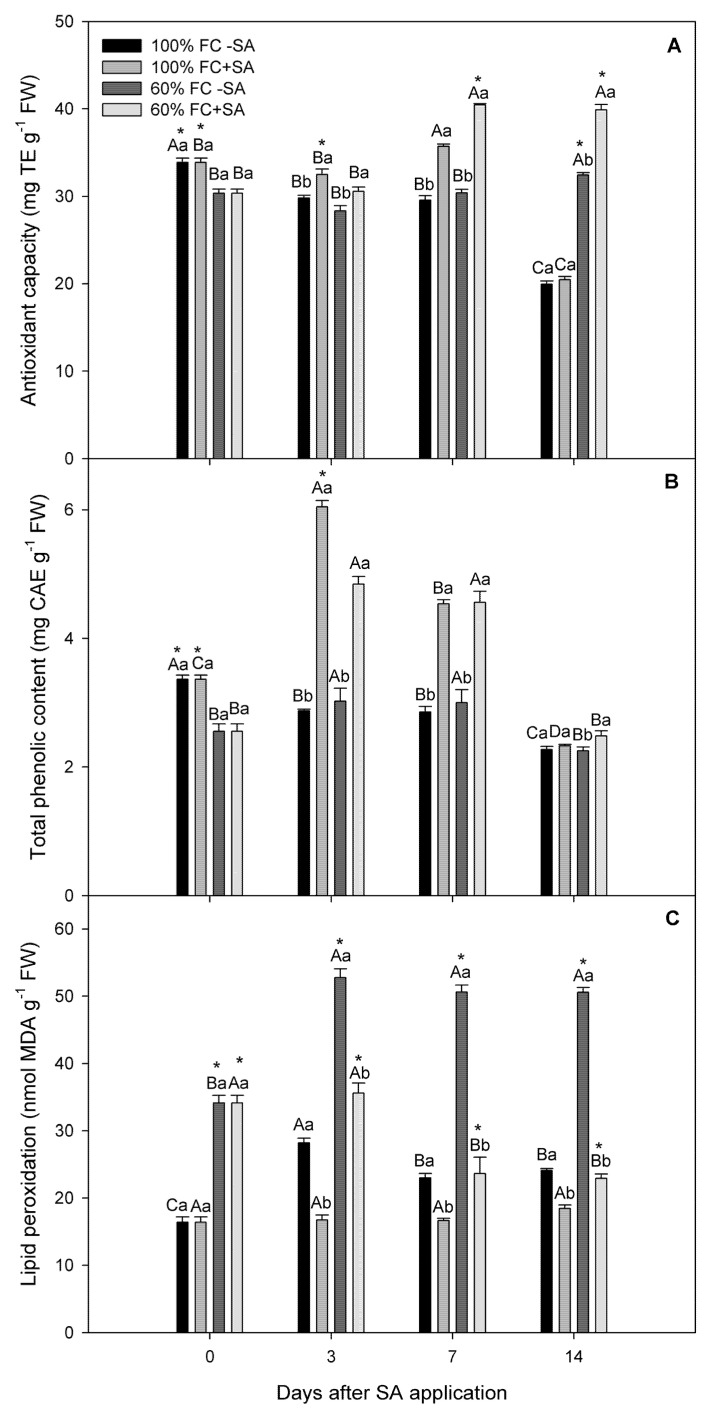
Non-enzymatic antioxidant system and oxidative stress in *A. chilensis* plants grown 100% and 60% FC and two SA doses (0 and 0.5 mM) at different times. (**A**) Antioxidant capacity, (**B**) total phenolic content, and (**C**) lipid peroxidation. The bars are means ± SE (*n* = 3). Different uppercase letters indicate significant differences among times post SA application for the same SA and irrigation treatment according to Tukey’s test (*p* ≤ 0.05). Different lowercase letters indicate significant differences between SA treatment for the same irrigation treatment and time post SA application according to Tukey’s test (*p* ≤ 0.05). According to Tukey’s test, asterisks indicate significant differences between irrigation treatments of the experiment for the same SA treatment and time post SA application (*p* ≤ 0.05).

**Figure 5 plants-11-00639-f005:**
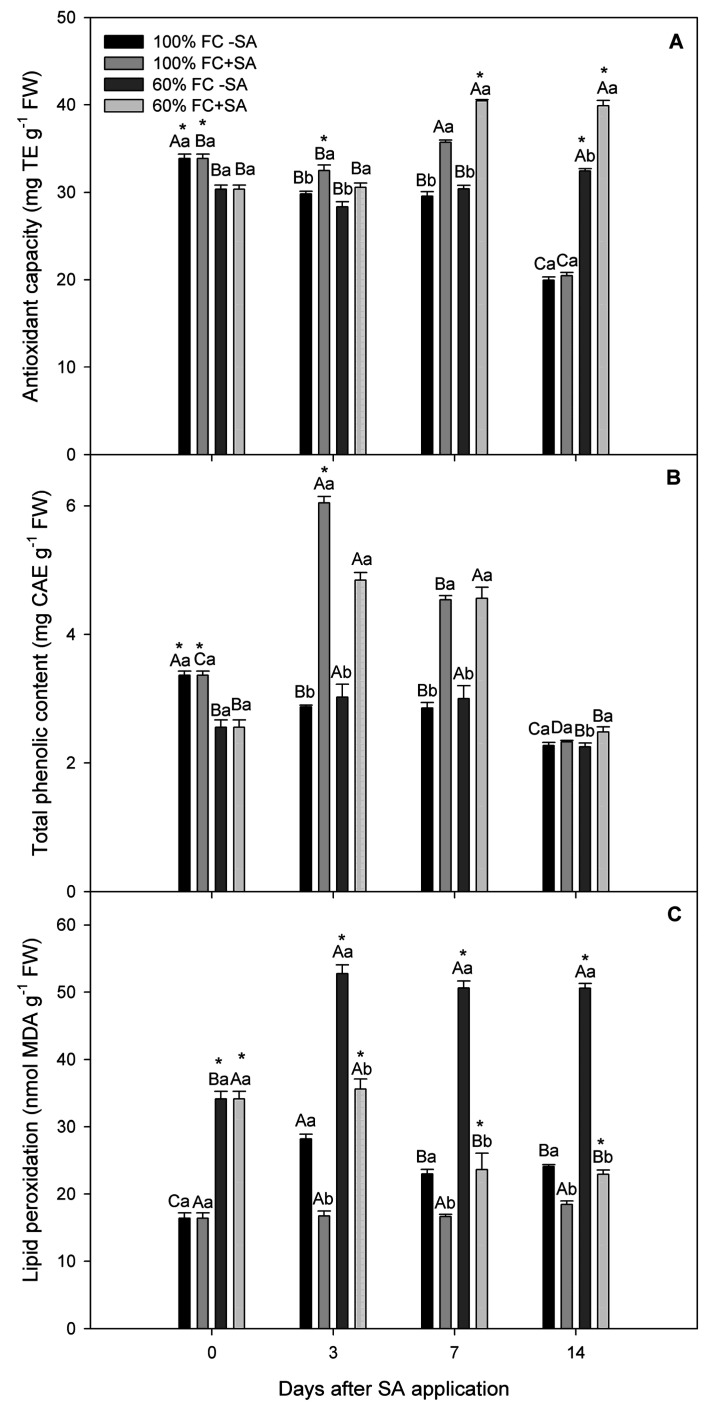
Enzymatic antioxidant system in *A. chilensis* plants grown under 100% and 60% FC and two SA doses (0 and 0.5 mM) at different times after SA application. (**A**) Superoxide dismutase (SOD) and (**B**) ascorbate peroxidase (APX) activity. The bars are means ± SE (*n* = 3). Different uppercase letters indicate significant differences among times post SA application for the same SA and irrigation treatment according to Tukey’s test (*p* ≤ 0.05). Different lowercase letters indicate significant differences between SA treatment for the same irrigation treatment and time post SA application according to Tukey’s test (*p* ≤ 0.05). According to Tukey’s test, asterisks indicate significant differences between irrigation treatments of the experiment for the same SA treatment and time post SA application (*p* ≤ 0.05).

**Figure 6 plants-11-00639-f006:**
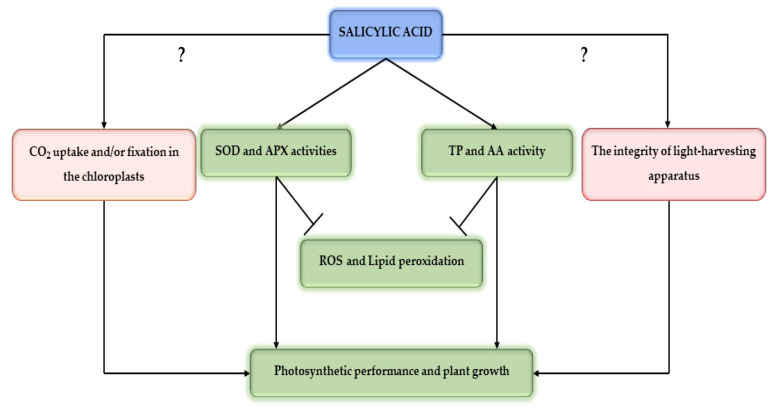
Schematic representation of SA-induced mechanism to tolerate drought stress in *A. chilensis* plants. We suggested that SA induces enzymatic and non-enzymatic antioxidant mechanisms to tolerate drought stress, decrease oxidative stress, and improve photosynthesis performance and plant growth in *A. chilensis* plants. Abbreviations: superoxide dismutase (SOD), ascorbate peroxidase (APX), total phenolic content (TPC), antioxidant capacity (AC), reactive oxygen species (ROS).

**Table 1 plants-11-00639-t001:** Relative growth rates of *A. chilensis* plants grown under 100% and 60% FC and two SA doses (0 and 0.5 mM) at different times post SA application.

Relative Growth Rate (mg Dry Weight Day^−1^)			
Treatment	Day 3	Day 7	Day 14
100% FC-SA	47.14 ± 1.14 Ab *	45.96 ± 0.39 Ab *	45.71 ± 2.34 Aa *
100% FC + SA	52.08 ± 1.29 Aa *	52.90 ± 1.69 Aa *	44.70 ± 2.85 Ba *
60% FC-SA	37.87 ± 2.18 Ab	38.54 ± 1.07 Ab	37.65 ± 0.77 Ab
60% FC + SA	42.70 ± 1.43 Aa	41.21 ± 1.09 Aa	39.63 ± 0.65 Aa

The bars are means ± SE (*n* = 3). Different uppercase letters indicate significant differences among post SA application time for the same SA and irrigation treatment according to Tukey’s test (*p* ≤ 0.05). Different lowercase letters indicate significant differences between SA treatment for the same irrigation treatment and day according to Tukey’s test (*p* ≤ 0.05). According to Tukey’s test, asterisks indicate significant differences between irrigation treatments of the experiment for the same SA treatment and time post SA application (*p* ≤ 0.05).

## Data Availability

The data presented in this study are available in the results section.

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
