# Peer review of "Salicylic Acid Improves Antioxidant Defense System and Photosynthetic Performance in Aristotelia chilensis Plants Subjected to Moderate Drought Stress"

_plants, 2022, doi:10.3390/plants11050639_

Round 1

Reviewer 1 Report

At first, I was concerned that 60% field capacity could not be enough drought stress to affect the performance of this plant and answer the question proposed in the study, that is to test whether a salicylic acid addition can help out the plant to cope with the effects of drought stress: oxidative stress and reduced photosynthetic capacity. After seeing the leaf water potential and leaf relative water content data from Figure 1 (the black filled symbol series), it seems that the moderate drought stress is well performed. Also lipid peroxidation is high (*) in the stressed plants (Figure 4a), and the photosynthesis-related variables (Figure 3) at day 0. The manuscript seems well written and the conclusions supported by the results and statistical tests.

-L413-l416. The measurement of leaf water potnetial is wrong, after my long experience with pressure chambers. As I understand, the authors measured leaf water potential with a Scholander pressure chamber in the morning, after having covered the leaves with aluminium foil in a plastic bag. I understand that the leaves were still attached to the branch (the authors do not say but seem logical in the procedure). What the authors are doing is following a protocol for estimating the water potential of the xylem by preventing leaf transpiration. Please, refer to Navarro et al. 2018, GCB-Bioenergy (https://doi.org/10.1111/gcbb.12526):

"For the determination of Ψx, leaves were covered with both a plastic bag and aluminum foil for at least 2 hr before the measurement. Bagging prevented leaf transpiration, allowing the leaf water potential to equal the stem water potential (Begg & Turner, 1970)."

- The authors cite Matthews et al 1981 (https://doi.org/10.5073/vitis.1987.26.147-160), which at the end of page 148 describes how to measure water potential at midday, by covering a leaf with a plastic bag and immediately transferring to the pressure chamber. The authors should use a more appropriate reference, if they did not follow this protocol.

- Midday water potential is a more typical measure than morning water potential. Why did the authors choose to measure in the morning?

-L265 and Figure 1: MegaPascal is written "MPa"

- L397. Why did the authors choose one concentration (or one dose level) of salicylic acid? Please, clarify this with references.

- The authors justify that 60% is a moderate stress level for this plant (L396), but I can't find where the authors retrieved the protocol for fumigation with SA.

- And why not more than one level of drought stress? It seems risky to have bet it all to only one drought level.

-L275-L278, L306-L310, and L313-L315 seem like 'Results'

Author Response

Please see the attached response letter 

Reviewer 2 Report

It is opinion of the reviewer that this paper before acceptance needs several revisions. My individual comments are listed below.

26, 104, 266 – It should be “MPa”.

35 – It should be “phenolics”.

Throughout entire manuscript, the authors should use a term of “antioxidant enzymes” instead of “enzymatic enzymes”.

64 – Some phenolic compounds can inhibit lipoxygenase activity.

191– It should be “total phenolic content”.

191 – The term of “antioxidant activity” is used for the extracts or pure compounds. For plants, a properly is a term of “potential” or “capacity”.

397 – It is not clear if authors used 0.5 mol SA or the volume of 0.5 mol/L SA (mM = mmol/L).

430 – “ppm’ is not any unit of SI.

436 – Remove “spectrophotometrically”.

436 – The lipid extraction for TBARS should be briefly described.

437 – How the method was calibrated? Equation should be reported.

441 – How the extract of phenolic compounds was obtained?

441 – It should be “DPPH (2.2-diphenyl-1-picryl-441 hydrazyl) assay”

443 – It should be “equivalents”.

447 – How the results were expressed?

452 – How the crude extract was obtained?

463 – What does it mean “absorbance decomposition”?

463 – It should be “crude” instead of “coarse”.

463 – How the crude extract was obtained?

473 – It should be “Tukey’s”.

599, 603 – It should be “PLoS ONE”.

Author Response

Please see the attached response letter.

Round 2

Reviewer 1 Report

I would like to thank the authors for having taken in account my comments and suggestions. The corrected manuscript seems to be ready for publication.
Cordially

Author Response

Thank for all valuable comments

Reviewer 2 Report

The authors corrected this paper properly taken under considerations all my comments. Therefore, I can accept it now.

Author Response

Thank for all valuable comments